# Effects of simulated reduced gravity and walking speed on ankle, knee, and hip quasi-stiffness in overground walking

**Mhairi K. MacLean**[1]\*, **Daniel P. Ferris**[2]

**1** Department of Biomechanical Engineering, University of Twente, Enschede, The Netherlands, **2** J. Crayton Pruitt Family Department of Biomedical Engineering, University of Florida, Gainesville, Florida, United States of America

\* M.K.MacLean@UTwente.nl

**Data Availability Statement:** The data has been uploaded to figshare. It has been added to my outstanding collection of data from the Overground Bodyweight Supported study. The collection is stored at https://figshare.com/collections/

## Abstract

Quasi-stiffness characterizes the dynamics of a joint in specific sections of stance-phase and is used in the design of wearable devices to assist walking. We sought to investigate the effect of simulated reduced gravity and walking speed on quasi-stiffness of the hip, knee, and ankle in overground walking. 12 participants walked at 0.4, 0.8, 1.2, and 1.6 m/s in 1, 0.76, 0.54, and 0.31 gravity. We defined 11 delimiting points in stance phase (4 each for the ankle and hip, 3 for the knee) and calculated the quasi-stiffness for 4 phases for both the hip and ankle, and 2 phases for the knee. The $R^2$ value quantified the suitability of the quasi-stiffness models. We found gravity level had a significant effect on 6 phases of quasi-stiffness, while speed significantly affected the quasi-stiffness in 5 phases. We concluded that the intrinsic muscle-tendon unit stiffness was the biggest determinant of quasi-stiffness. Speed had a significant effect on the $R^2$ of all phases of quasi-stiffness. Slow walking (0.4 m/s) was the least accurately modelled walking speed. Our findings showed adaptions in gait strategy when relative power and strength of the joints were increased in low gravity, which has implications for prosthesis and exoskeleton design.

## Introduction

Quantifying the quasi-stiffness of the human lower limb joints during locomotion can provide insight to gait biomechanics and be valuable for the design of robotic exoskeletons and prostheses. Quasi-stiffness, sometimes referred to as dynamic stiffness, is a linear model of the relationship between joint moment and angle during particular phases of the gait cycle [1,2]. The joint is modelled as a torsional spring with the external or negative internal joint moment proportional to the joint angle. Past studies have quantified joint quasi-stiffness for the hip, knee, and ankle during human walking [2–6]. Quasi-stiffness during the power absorption phase of the biological joint can provide an indication of potential absorbed energy that may be recovered later in the gait cycle. Ankle quasi-stiffness has been the most investigated of the lower limb joints with findings showing the quasi-stiffness changes with gait speed, added mass, and bodyweight [5–8]. Studies have also shown changes in quasi-stiffness under external perturbations [9].

Overground_walking_with_simulated_reduced_
gravity/5430234 (DOI: https://doi.org/10.6084/m9.
figshare.c.5430234.v2) The subject and trial
individual data is stored at https://figshare.com/
articles/dataset/Individual_Quasi-Stiffness_Data/
20340192 (DOI: 10.6084/m9.figshare.20340192)
The subject averaged data is stored at https://
figshare.com/articles/dataset/Final_Averaged_
Quasi-stiffness_Data/20338971 (DOI: 10.6084/m9.
figshare.20338971)

**Funding:** The research was partially funded by
National Institutes of Neurological Disorders and
Stroke, grant number R01NS104772 (DPF). The
funders had no role in study design, data collection
and analysis, decision to publish, or preparation of
the manuscript. Daniel Ferris received the award.

**Competing interests:** The authors have declared
that no competing interests exist.

Joint quasi-stiffness during human walking has been useful in control systems of robotic exoskeletons, prostheses, and bipedal robots which mimic the normal biomechanical behaviour of the human lower limb [10–12]. A common control method for wearable robotics is to set the relative impedances (e.g. stiffness and damping) and equilibrium points of the mechanical joints for different phases of the gait cycle. Biological quasi-stiffness provides information on suitable robot impedances. This technique of finite state machine control allows for smooth walking assistance without directly coupling the user's neural commands to the exoskeleton. For people who are unable to generate sufficient resistance to ground reaction forces, an exoskeleton/prosthesis with finite state machine controlled stiffness can provide appropriate resistance at a biological joint to prevent limb collapse during stance phase. Although joint quasi-stiffness is a simplified model of joint dynamics, it provides an approximation that has practical engineering applications.

Study of joint quasi-stiffness during different gait conditions may also provide insight into human neural control [13,14]. Joint quasi-stiffness is determined by the state of all passive and active tissues that cross the joint [15,16]. The biggest factor influencing joint quasi-stiffness is the stiffness of the muscle-tendon units actuating the joint. Muscle-tendon stiffness is comprised of the stiffness of the tendon and the stiffness of the muscles which attach to the tendon [17]. Joint quasi-stiffness should theoretically increase with muscle force and with increases in tendon stiffness that occur when a tendon is stretched. The former is an active mechanism and the latter a passive mechanism, but both are related to muscle force. A biological question of interest is if the preferred joint quasi-stiffness is dependent on muscle force. Although past studies have been able to examine quasi-stiffness over a small range of joint torques via changing walking speed and added mass, they have not studied a large range of peak torque values [7,18,19].

Used as a gait rehabilitation therapy, simulated reduced gravity reduces the joint torques and muscle forces required to produce gait [20,21]. One technique to simulate reduced gravity is to apply a constant upwards force to the torso with a harness, counteracting gravity without altering body mass [22]. The effect of gravity on the arms and legs when not in contact with the ground is unchanged. The reduced ground reaction forces from simulated reduced gravity can be very useful for gait rehabilitation, allowing people to practice walking if they normally struggle to walk independently. Supporting bodyweight contributes a large portion of the energetic cost of walking [22–25]. Reducing the effective bodyweight also decreases joint contact forces and net muscle moments during walking, essentially making it easier to walk [20,21,26]. In rehabilitation, simulated reduced gravity is often referred to as bodyweight support and has been used for gait rehabilitation of people with spinal cord injury, post-stroke hemiparesis, traumatic leg injury, and other neurological disabilities [27–32]. Practicing walking in simulated reduced gravity can improve muscle strength, limb and muscle coordination, and bone density [33,34]. To the best of our knowledge, there has been no investigation of joint quasi-stiffness in clinical populations in normal or reduced gravity. To parse out the effects and interactions of gravity level, walking speed, and underlying pathological physiology on quasi-stiffness, we first need an understanding of the relationship between gravity level, walking speed, and joint quasi-stiffness in a healthy population.

Biomechanists use simulated reduced gravity to test hypotheses about how gravity and bodyweight affect human movement. Past studies have shown that simulated reduced gravity alters the kinematic, kinetic, and electromyographic components of gait in able-bodied humans [20]. For example, the walk to run transition speed decreases, stance phase duration decreases, energetic cost decreases, joint contact forces and net muscle moments decrease, and electromyographic activity of some leg muscles decrease in stance phase [24,35–42]. These

papers have made it clear that the force due to gravity acting on the torso plays a large role in determining how humans control the motion of their bodies during locomotion.

The purpose of this study was to understand the effect of simulated reduced gravity on ankle, knee, and hip quasi-stiffness walking overground at a range of speeds. Simulated reduced gravity will reduce joint moments during the stance phase of walking. If there is a relationship between joint moment and quasi-stiffness, it should be noticeable with a three-fold range of gravity levels. We hypothesized that joint quasi-stiffness would decrease with simulated reduced gravity as it is likely dependent on the intrinsic muscle-tendon unit stiffness which scales with muscle and tendon force [15,43–45].

## Methods

### Data collection

12 able bodied individuals with no history of neuromuscular conditions participated in the study (6 female, 26±4 years old, body mass 70±8 kg, height 1.74±0.07 m, mean±s.d.). Participant demographics are detailed in S1 Table. Each participant signed an informed consent form approved by the University of Florida's institutional review board. The bodyweight support system used in this study (described in detail in [46]) uses constant force springs to provide a constant upwards force on the participant via a harness. Fluctuations in support force were less than ±5% bodyweight at normal gravity (1 G). Participants wore the harness and walked over an 8m overground walkway at 0.4, 0.8, 1.2, and 1.6 m/s in 1, 0.76, 0.54, and 0.31 gravity (G).

The walkway had 3 embedded force plates (AMTI, Watertown, MA, USA) to measure ground reaction forces at 1000 Hz and we used a visual motion capture system (Optitrack, Corvallis, OR, USA) to measure joint kinematics at 100 Hz. We palpated for and placed 22 reflective markers on the following anatomical landmarks: Ilium anterior superior, ilium posterior superior, greater trochanter of the femur, femoral lateral epicondyle, femoral medial epicondyle, fibula apex of the lateral malleolus, tibia apex of medial malleolus, the posterior of the calcaneus, heads of the 1st and 5th metatarsals, and the hallux. In addition to the anatomical markers, participants wore rigid bodies with 4 reflective markers on each shank and thigh. To control for walking speed, we used infra-red timing gates set-up before and after the force plates.

Participants completed all speed conditions for a level of simulated reduced gravity, rested for 5–15 minutes, then moved onto walking at the next level of reduced gravity. For each participant, we randomized the order of gravity levels and the order of the speed conditions for each simulated gravity level. To ensure the subjects' biomechanics adapted to simulated reduced gravity, participants practiced walking for 3 minutes at each gravity level before data were collected. Participants received no specific instructions on how to walk in reduced gravity, although we did request that they do their best to not include a flight phase in the highest walking speed and lowest gravity condition. We collected kinematic and kinetic data for 4 trials wherein the participant's right foot struck a single force plate, and the left foot landed on a subsequent force plate or straddled the remaining two force plates.

### Data processing

We attained joint angles and net internal moments from Visual 3D (C-Motion, Germantown, MD, USA) and determined quasi-stiffness with custom MATLAB (Mathworks, Natick, MA, USA) scripts. We first filtered motion capture and force plate data with a zero lag, 6 Hz lowpass 4[th] order Butterworth filter and then cut the data to stance phase (right foot initial contact to right foot toe off). An 18 N threshold on the vertical ground reaction force identified initial

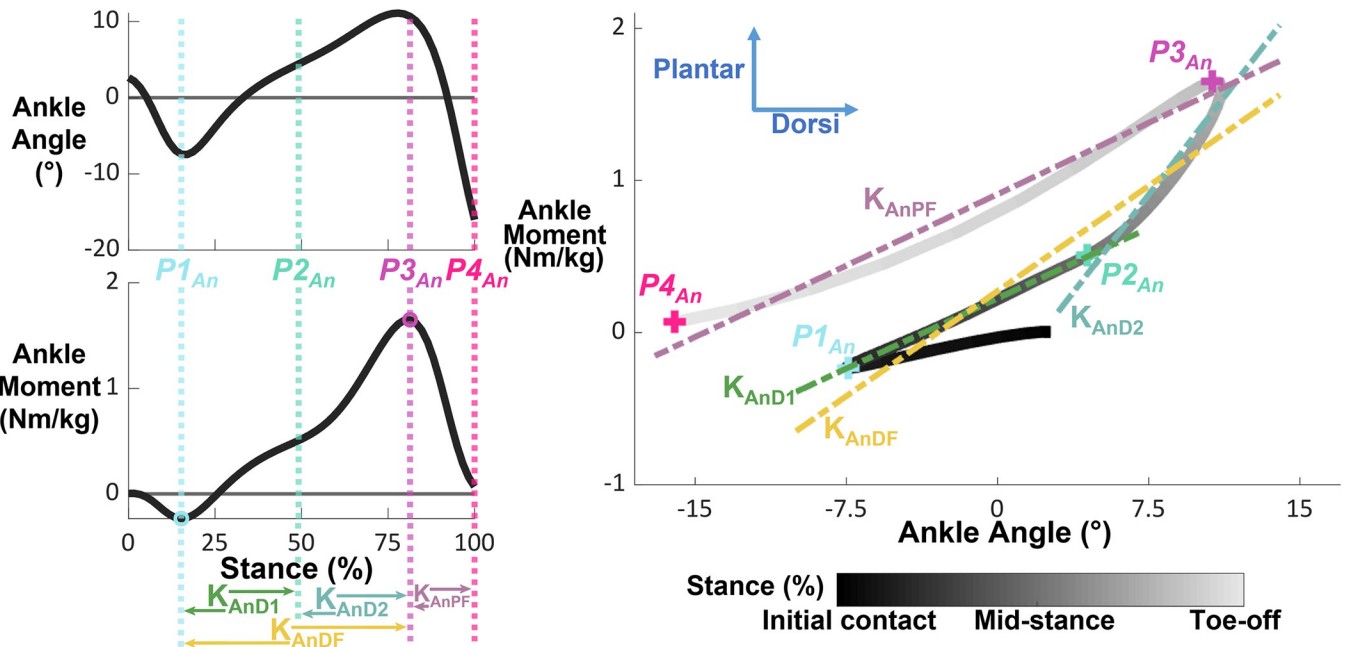

**Fig 1. Ankle angle, moment, and moment-angle relationships during stance phase overlaid with points of interest and phases of quasi-stiffness.** The smaller subplots on the left show ankle angle and moment during stance phase. The dashed vertical lines represent the timings of each point of interest, while the open circles identify the point of interest itself. The phases of quasi-stiffness are identified at the bottom of these plots, with the arrows indicating their onset and termination. The large, right figure shows the moment-angle relationship with the quasi-stiffness overlaid as dashed lines and the points of interests as "+". Data for visualization were normalized to 60 data points and averaged across all participants walking in normal gravity at 1.6 m/s. Direction of internal joint moment and joint angle are indicated with arrows on the rightmost plot.

contact and toe off. To build our anthropometric model for joint angle and moment calculations in Visual3D, we used default segment definitions and estimated the hip joint location with the Coda pelvis model [47]. To facilitate averaging across participants, we normalized the net internal moment to body mass. We will refer to normalized joint internal net moment as joint moment in this paper.

We defined and calculated 4 phases of quasi-stiffness for both the hip and ankle, and 2 phases of quasi-stiffness for the knee (Figs 1–3 and Table 1). We chose phases of quasi-stiffness delimited by points of interest (Table 2) with reference to previous studies [1–6,17]. We calculated the quasi-stiffness as the slope between joint angle (degrees) and normalized joint internal net moment (Nm/kg).

**Ankle.** The 4 points of interest for the ankle were the: minimum local ankle plantarflexion moment at the start of stance phase ($P1_{An}$), temporal mid-point of stance ($P2_{An}$), maximum plantar flexion moment ($P3_{An}$), and toe-off ($P4_{An}$). At normal (1.2 m/s) to fast (1.6 m/s) speeds, the mid-point of stance typically coincided with the low point of the "m"-shaped vertical ground reaction force and the point where the anterior-posterior force crossed from positive to negative. The quasi-stiffness phases defined by the consecutive points of interest were: early dorsi-flexion ($K_{AnD1}$) from $P1_{An}$ to $P2_{An}$, late dorsi-flexion ($K_{AnD2}$) from $P2_{An}$ to $P3_{An}$, and plantar-flexion ($K_{AnPF}$) from $P3_{An}$ to $P4_{An}$. We defined an additional ankle quasi-stiffness phase as total dorsi-flexion ($K_{AnDF}$) which we found between $P1_{An}$ and $P3_{An}$. We did not calculate quasi-stiffness in the early and late dorsi-flexion phases if the temporal mid-point of stance ($P2_{An}$) occurred after the point of maximum plantar-flexion moment ($P3_{An}$). If the duration between $P2_{An}$ and $P3_{An}$ was less than 0.05 s, we did not calculate quasi-stiffness of the

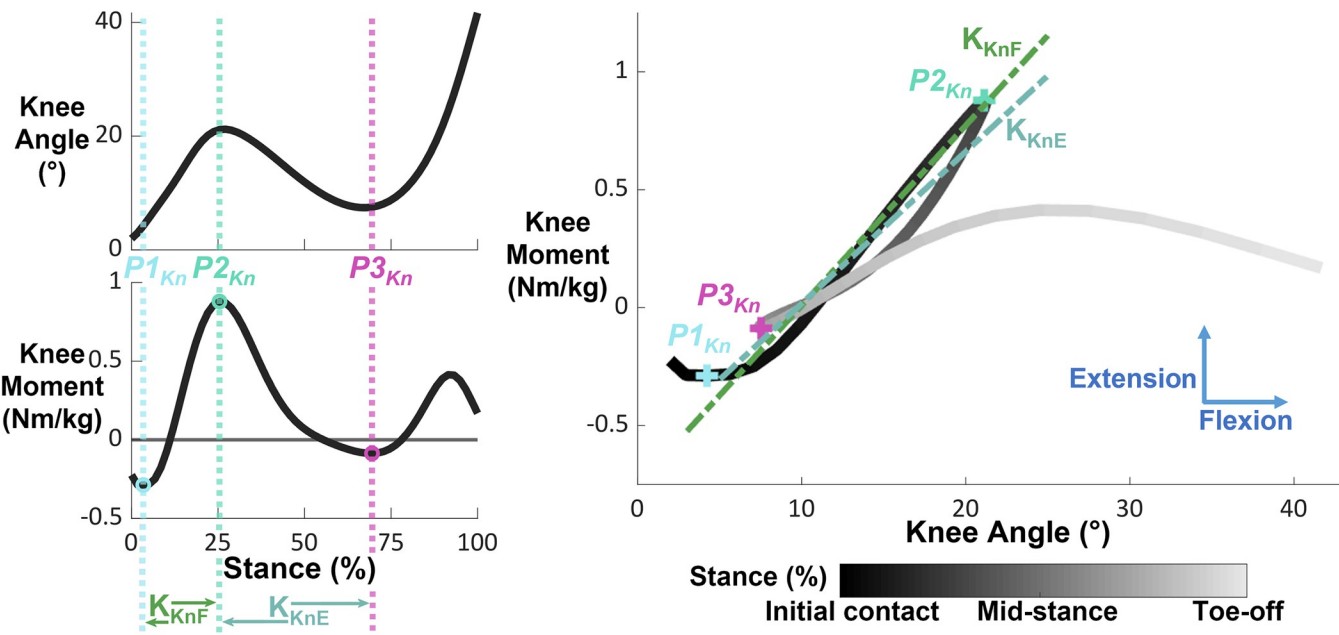

**Fig 2. Knee angle, moment, and moment-angle relationships during stance phase overlaid with points of interest and phases of quasi-stiffness.** The smaller subplots on the left show knee angle and moment during stance phase. The dashed vertical lines represent the timings of each point of interest, while the open circles identify the point of interest itself. The phases of quasi-stiffness are identified at the bottom of these plots, with the arrows indicating their onset and termination. The large, right figure shows the moment-angle relationship with the quasi-stiffness overlayed as dashed lines and the points of interests as "+". Data for visualization were normalized to 60 data points and averaged across all participants walking in normal gravity at 1.6 m/s. Direction of internal joint moment and joint angle are indicated with arrows on the rightmost plot.

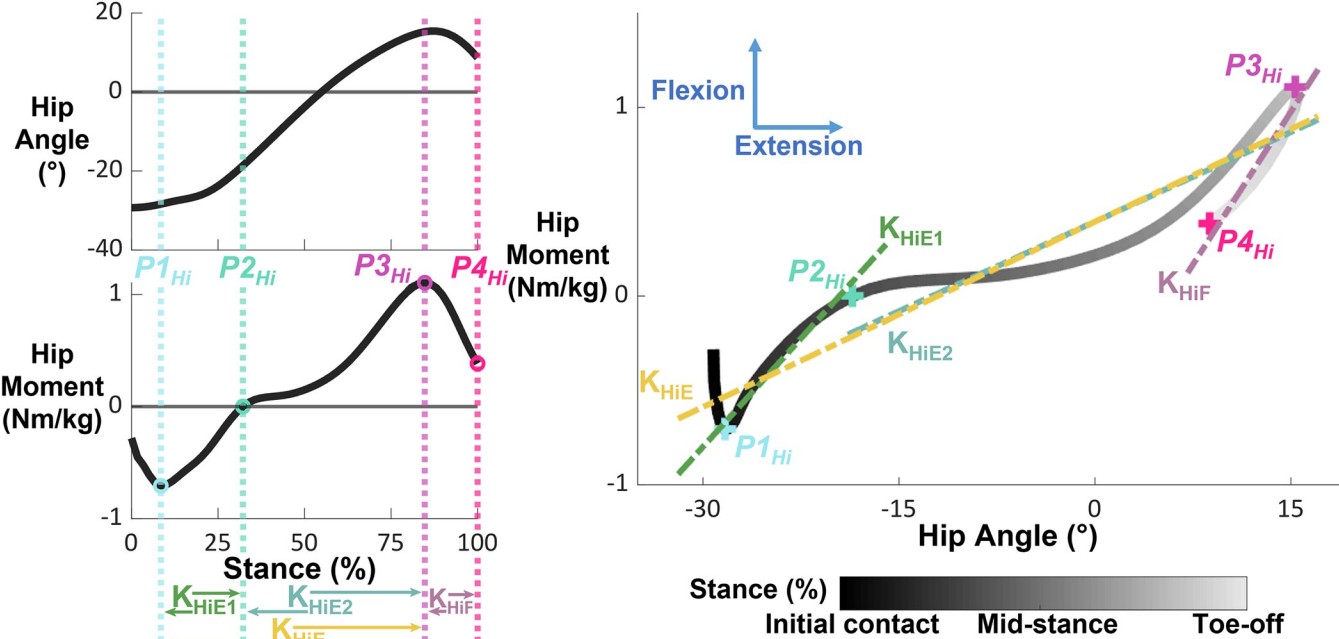

**Fig 3. Hip angle, moment, and moment-angle relationships during stance phase overlaid with points of interest and phases of quasi-stiffness.** The smaller subplots on the left show hip angle and moment during stance phase. The dashed vertical lines represent the timings of each point of interest, while the open circles identify the point of interest itself. The phases of quasi-stiffness are identified at the bottom of these plots, with the arrows indicating their onset and termination. The large, right figure shows the moment-angle relationship with the quasi-stiffness overlayed as dashed lines and the points of interests as "+". Data for visualization were normalized to 60 data points and averaged across all participants walking in normal gravity at 1.6 m/s. Direction of internal joint moment and joint angle are indicated with arrows on the rightmost plot.

**Table 1. All quasi-stiffness phases, their acronym, and the points of interest between which they were found.**

| | Quasi-Stiffness | | Point of Interest | |
| --- | --- | --- | --- | --- |
| | | | Starting | Ending |
| Ankle | Early dorsi-flexion | $K_{AnD1}$ | $P1_{An}$ | $P2_{An}$ |
| | Late dorsi-flexion | $K_{AnD2}$ | $P2_{An}$ | $P3_{An}$ |
| | Total dorsi-flexion | $K_{AnDF}$ | $P1_{An}$ | $P3_{An}$ |
| | Plantar-flexion | $K_{AnPF}$ | $P3_{An}$ | $P4_{An}$ |
| Knee | Knee flexion | $K_{KnF}$ | $P1_{Kn}$ | $P2_{Kn}$ |
| | Knee extension | $K_{KnE}$ | $P2_{Kn}$ | $P3_{Kn}$ |
| Hip | Early hip extension | $K_{HiE1}$ | $P1_{Hi}$ | $P2_{Hi}$ |
| | Late hip extension | $K_{HiE2}$ | $P2_{Hi}$ | $P3_{Hi}$ |
| | Total hip extension | $K_{HiE}$ | $P1_{Hi}$ | $P3_{Hi}$ |
| | Hip flexion | $K_{HiF}$ | $P3_{Hi}$ | $P4_{Hi}$ |

late dorsi-flexion phase for that trial. Fig 1 visualizes the points of interest and quasi-stiffness phases for the ankle.

**Knee.** For the knee, we defined the 3 points of interest as the: local minimum knee extension moment at the beginning of stance ($P1_{Kn}$), maximum knee extension moment ($P2_{Kn}$), and local minimum knee extension moment occurring after $P2_{Kn}$ ($P3_{Kn}$). We calculated the quasi-stiffness phases of knee flexion ($K_{KnF}$) and knee extension ($K_{KnE}$) between $P1_{Kn}$ and $P2_{Kn}$, and $P2_{Kn}$ and $P3_{Kn}$ respectively. The points of interest and quasi-stiffness phases of the knee are displayed in Fig 2.

**Hip.** The hip moment-angle relationship had 4 points of interest: local minimum hip flexion moment at the beginning of stance ($P1_{Hi}$), when the hip moment transitioned from extension to flexion ($P2_{Hi}$), local maximum hip flexion moment occurring near the end of stance ($P3_{Hi}$), and local minimum hip flexion moment before toe-off, or if there was no local minimum, toe-off ($P4_{Hi}$). At normal to fast speeds (1.2 & 1.6 m/s) in normal gravity, the transition of hip moment from extension to flexion coincided with a local maximum hip flexion moment before the temporal mid-point of stance. We defined the 3 consecutive quasi-stiffness phases of the hip as: early extension ($K_{HiE1}$) between $P1_{Hi}$ and $P2_{Hi}$, late extension ($K_{HiE2}$) between $P2_{Hi}$ and $P3_{Hi}$, and flexion ($K_{HiF}$) between $P3_{Hi}$ and $P4_{Hi}$. Between $P1_{Hi}$ and $P3_{Hi}$, we defined a $4^{th}$ phase of quasi-stiffness ($K_{HiE}$) encapsulating the total phase of hip extension. In trials where the hip moment did not transition from extension to flexion, we did not calculate $K_{HiE1}$

**Table 2. All points of interest for finding quasi-stiffness phases and their description.**

| Point of interest | Descriptor |
| --- | --- |
| $P1_{An}$ | Minimum local ankle plantarflexion moment (in early stance) |
| $P2_{An}$ | Temporal mid-point of stance |
| $P3_{An}$ | Maximum plantar flexion moment |
| $P4_{An}$ | Toe-off |
| $P1_{Kn}$ | Local minimum knee extension moment (in early stance) |
| $P2_{Kn}$ | Maximum knee extension moment |
| $P3_{Kn}$ | Local minimum knee extension moment (after $P2_{Kn}$) |
| $P1_{Hi}$ | Local minimum hip flexion moment (in early stance) |
| $P2_{Hi}$ | Transition of hip moment from extension to flexion |
| $P3_{Hi}$ | Local maximum hip flexion moment (in late stance) |
| $P4_{Hi}$ | Local minimum hip flexion moment before toe-off (if no local minimum, toe-off was used) |

and $K_{HiE2}$. If the time between 2 consecutive points of interest was less than 0.05 seconds, we did not calculate quasi-stiffness for that phase of gait. Fig 3 illustrates the points of interest and quasi-stiffness phases of the hip.

## Calculating quasi-stiffness

We found the 11 points of interest for each walking trial, then calculated the stiffness of each quasi-stiffness phase using a least squares linear model. For each walking trial, we applied the linear model as a function of joint angle (i.e. predicted joint moment from given joint angles). To determine the fit of the model for each phase of quasi-stiffness in each trial, we calculated the $R^2$ value. We also calculated the duration of each phase of quasi-stiffness as a percentage of the total stance time. Finally, we averaged quasi-stiffness values, $R^2$, and phase durations across each walking trial in a condition for every participant, and then averaged across participants.

## Statistics

To evaluate if speed and gravity had a significant effect on quasi-stiffness and quality of fit, we used a linear mixed model with simulated gravity level and speed as repeated factors, and participants as a random factor. We performed the statistical tests with SPSS (IBM Corp. Armonk, NY, USA). Due to data missing in our final dataset, a mixed linear model was deemed more suitable than an ANOVA. We used the Benjamini-Hochberg technique to make post-hoc comparisons (significance value of 0.05) [48,49].

## Results

The moment-angle loops for the hip, knee, and ankle changed shape with both level of reduced gravity and speed (Fig 4). In some walking conditions, particularly at the slower speeds (0.4 & 0.8 m/s) or high levels of reduced gravity, certain points of interest were not present in the moment-angle relationship. For example, 2 participants exhibited no hip flexion moment during the stance phase of walking in 0.31 G at 0.4 m/s, so $P2_{Hi}$ did not occur. Furthermore, we did not use the data of 2 participants in the 0.31 G, 1.6 m/s condition as they were unable to walk without a flight phase. Additionally, data were lost for 1 participant at normal gravity, 1.2 m/s walking. S2 Table indicates the number of participants data from which we calculated each quasi-stiffness phase for every walking condition. Table 3 reports the significance of gravity level and speed on quasi-stiffness values and quality of the fit from the linear mixed model. We list the mean quasi-stiffness and $R^2$ values in S3 and S4 Tables respectively.

## Ankle

Ankle quasi-stiffness tended to decrease with reduced gravity level (Fig 5). Level of simulated gravity significantly reduced quasi-stiffness of late dorsi-flexion ($K_{AnD2}$), total dorsiflexion ($K_{AnDF}$), and plantar-flexion ($K_{AnPF}$) (all p < 0.001). The quality of fit for $K_{AnD2}$ and $K_{AnDF}$ was significantly reduced with gravity level (p<0.001). We found a significant effect of speed on $K_{AnPF}$ (p<0.001) and walking speed was a significant factor for the quality of fit for all 4 linear models of stiffness (all p<0.001). The effect of speed on fit was not consistent across quasi-stiffness phases and gravity levels. Gravity level and speed appeared to impact the duration spent in each phase of quasi-stiffness. Duration of late dorsi-flexion and all dorsi-flexion tended to decrease with reduced gravity and increased speed. Plantar-flexion duration tended to increase with reduced gravity and slower speeds.

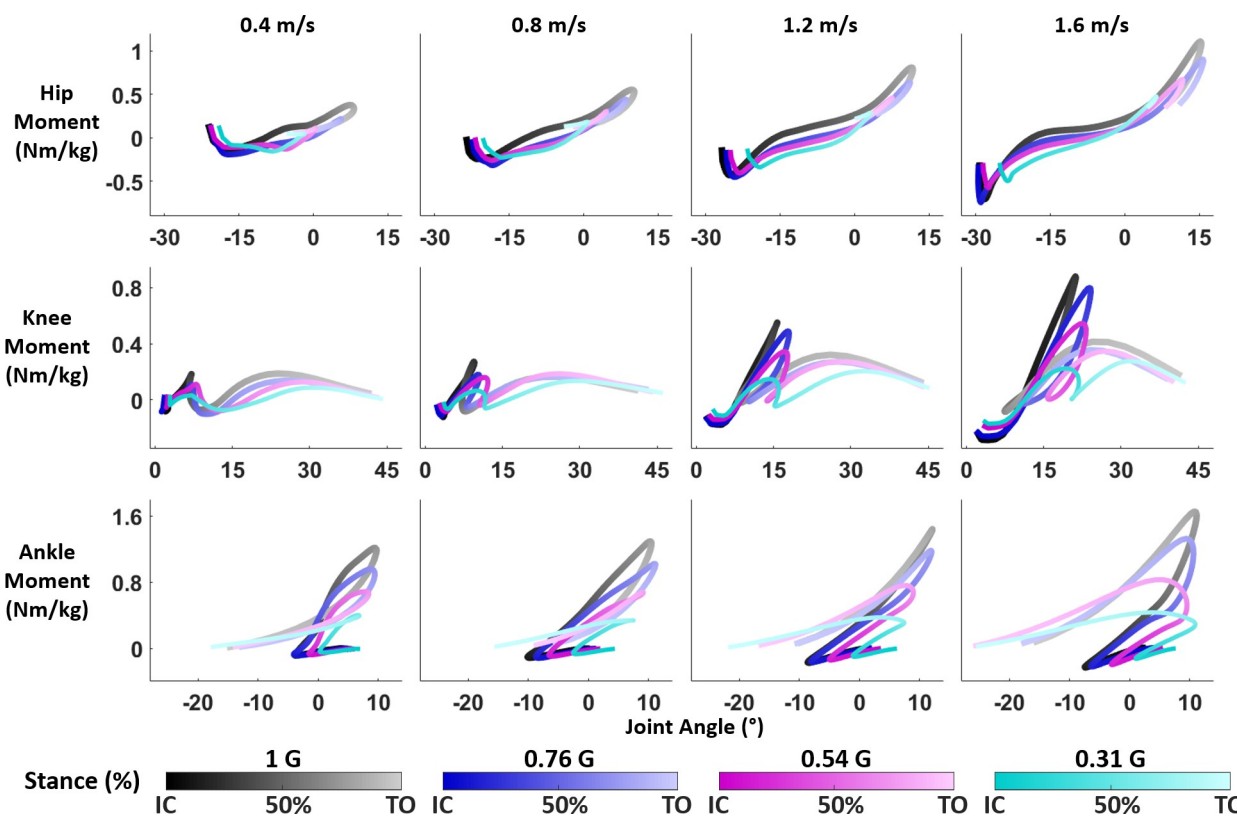

**Fig 4. Moment-angle loops in stance phase for the hip, knee, and ankle walking in 1, 0.76, 0.54, and 0.31 G at 0.4, 0.8, 1.2, and 1.6 m/s.** Data for visualization were normalized to 60 data points and averaged across all participants in each condition. Every x-axis displays joint angle of the respective joint.

## Knee

Simulated reduced gravity significantly decreased the quasi-stiffness of the knee in the flexion phase ($K_{KnF}$) across speeds (p<0.001), but had no significant effect on quasi-stiffness in the knee extension phase ($K_{KnF}$) (p>0.99) (Fig 6). Gravity level had no significant effect on the $R^2$ fits of the models. Duration of both flexion and extension stages tended to decrease with reduced gravity. Speed had a significant effect on the quasi-stiffness values and fits for knee flexion and extension (all p<0.001). There were no instances where $P1_{Kn}$, $P2_{Kn}$, or $P3_{Kn}$ could not be identified.

## Hip

Reduced gravity significantly reduced the quasi-stiffness of the hip in the early extension ($K_{HiE1}$) and the flexion stage ($K_{HiF}$) (p = 0.004 and p<0.001 respectively) (Fig 7). Reduced gravity also significantly impacted the fit of the quasi-stiffness model in the late phase of hip extension (p = 0.03). Increased walking speed significantly increased $K_{HiE1}$ and $K_{HiF}$ (all p<0.001). The fit of the linear model was significantly affected by walking speed for all 4 phases of quasi-stiffness (p<0.001 for $R^2$ of $K_{HiE1}$, $K_{HiF}$, and $K_{HiE}$. p = 0.02 for $R^2$ of $K_{HiE2}$). Percent of stance spent in each quasi-stiffness phase appeared to be affected by both gravity and speed. The proportion of time spent in the early hip extension phase increased with reduced gravity, while the late phase of hip extension was decreased. Reduced gravity also decreased the proportion of time spent in the hip flexion phase. Faster walking speeds tended

**Table 3. Post-hoc significance of gravity level and speed on quasi-stiffness and fits.**

| Joint | Quasi-stiffness | P value | | Quality of fit | P value | |
|---|---|---|---|---|---|---|
| | | Gravity | Speed | | Gravity | Speed |
| Ankle | $K_{AnD1}$ | >0.99 | >0.99 | $R^2$ of $K_{AnD1}$ | 0.741 | **<0.001*** |
| | $K_{AnD2}$ | **<0.001*** | >0.99 | $R^2$ of $K_{AnD2}$ | **<0.001*** | **<0.001*** |
| | $K_{AnDF}$ | **<0.001*** | >0.99 | $R^2$ of $K_{AnDF}$ | **<0.001*** | **<0.001*** |
| | $K_{AnPF}$ | **<0.001*** | **<0.001*** | $R^2$ of $K_{AnPF}$ | >0.99 | **<0.001*** |
| Knee | $K_{KnF}$ | **<0.001*** | **<0.001*** | $R^2$ of $K_{KnF}$ | >0.99 | **<0.001*** |
| | $K_{KnE}$ | >0.99 | **<0.001*** | $R^2$ of $K_{KnE}$ | 0.837 | **<0.001*** |
| Hip | $K_{HiE1}$ | **0.004*** | **<0.001*** | $R^2$ of $K_{HiE1}$ | >0.99 | **<0.001*** |
| | $K_{HiE2}$ | >0.99 | >0.99 | $R^2$ of $K_{HiE2}$ | **0.030*** | **0.020*** |
| | $K_{HiE}$ | >0.99 | >0.99 | $R^2$ of $K_{HiE}$ | >0.99 | **<0.001*** |
| | $K_{HiF}$ | **0.001*** | **<0.001*** | $R^2$ of $K_{HiF}$ | 0.332 | **<0.001*** |

Significant results are bolded and denoted with an asterisk.

to increase time spent in the late phase of hip extension and in the phase of hip flexion. The quasi-stiffness of hip flexion was often not calculable at 0.31 G as either i) the point of maximum hip flexion moment ($P3_{Hi}$) occurred at toe-off ($P4_{Hi}$), or ii) the time between maximum hip flexion moment ($P3_{Hi}$) and local minimum flexion moment ($P4_{Hi}$) was less than 0.05 s.

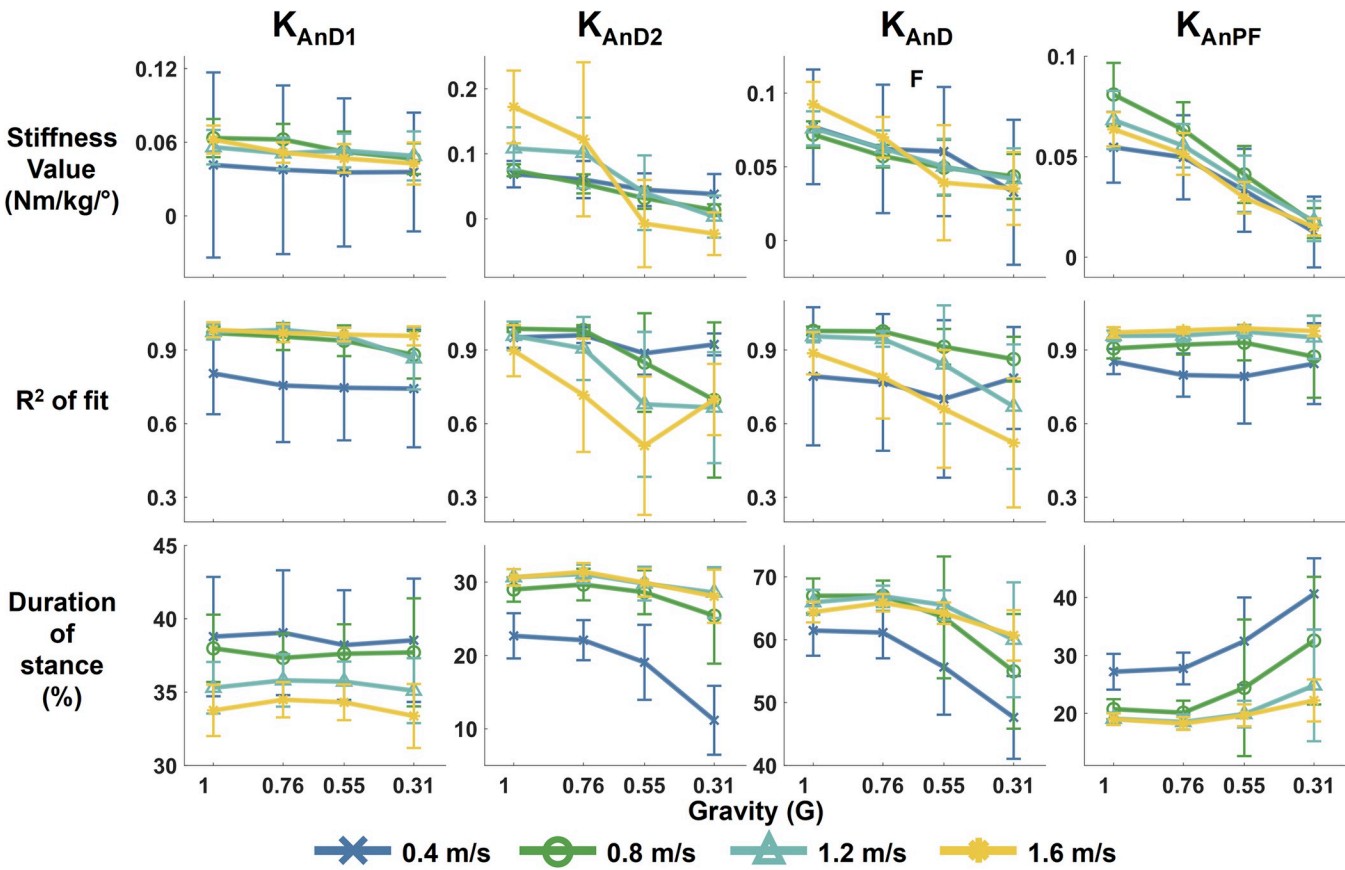

**Fig 5. Quasi-stiffness of the ankle in stance phase at different levels of simulated reduced gravity and speeds.** Top row shows stiffness values, middle row is the quality of fit of the linear quasi-stiffness model, and bottom row is the duration of the quasi-stiffness phase as percent of stance phase. Each column of plots is a different quasi-stiffness phase for the ankle: $K_{AnD1}$ is early dorsi-flexion, $K_{AnD2}$ is Ankle late dorsi-flexion, $K_{AnDF}$ is overall Ankle dorsi-flexion, and $K_{AnPF}$ is Ankle plantar-flexion. The error bars represent standard deviation.

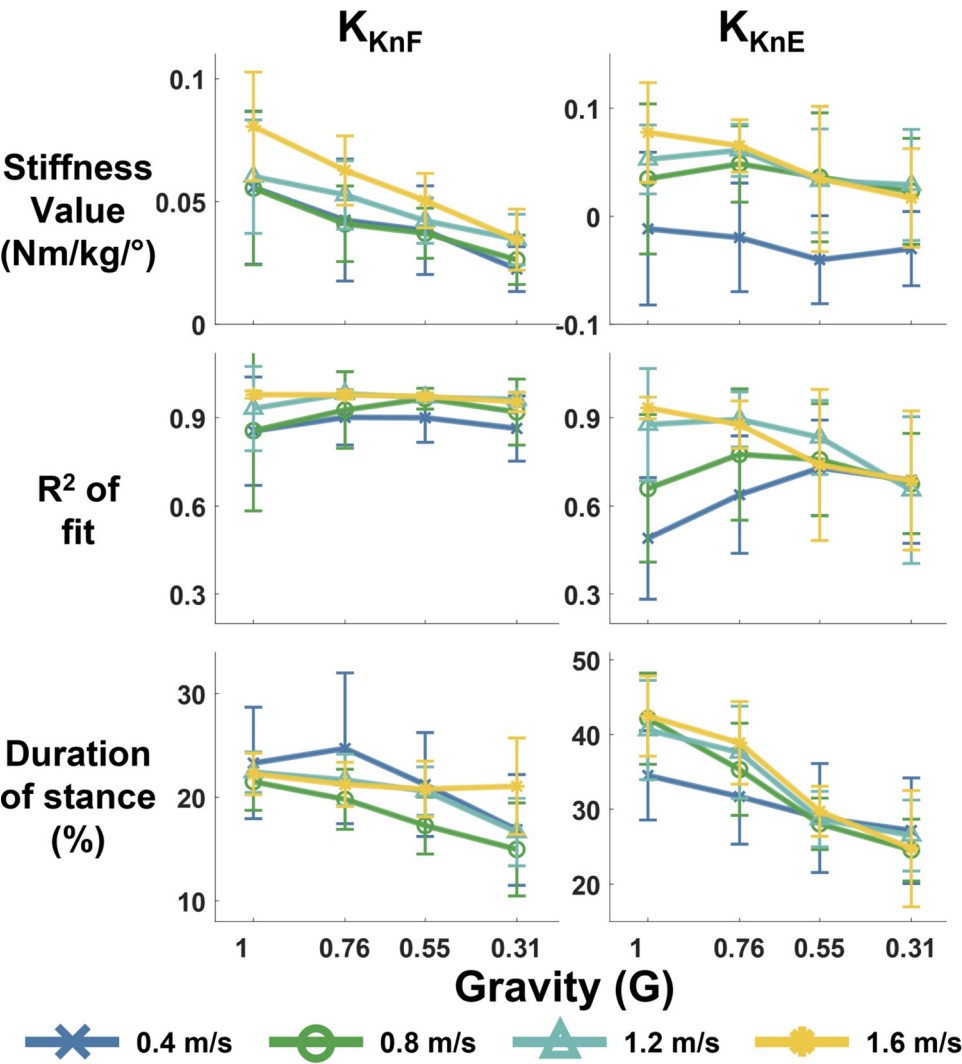

**Fig 6. Quasi-stiffness of the knee in stance phase at different levels of simulated reduced gravity and speeds.** Top row shows stiffness values, middle row is the quality of fit of the linear quasi-stiffness model, and bottom row is the duration of the quasi-stiffness phase as percent of stance phase. Each column of plots is a different quasi-stiffness phase for the knee: $K_{KnF}$ is flexion, and $K_{KnE}$ is knee extension. The error bars show standard deviation.

## Discussion

Simulated reduced gravity substantially modified joint moment-angle relationships in stance phase, prompting changes in quasi-stiffness values and linearity. In agreement with our hypothesis, artificially reducing gravity generally decreased the quasi-stiffness of the hip, knee, and ankle. Speed was a significant factor in determining the $R^2$ fit of the linear quasi-stiffness models, while gravity level had a smaller effect on $R^2$. Our results confirmed previous research which found speed to cause morphological changes in the moment-angle relationship of the ankle [7] and to be a predictor of stiffness [3–5].

### Biomechanical principles underlying joint quasi-stiffness

Joint quasi-stiffness tended to decrease with gravity level, which is in agreement with the supposition that quasi-stiffness is modulated by the inherent properties of the muscle-tendon

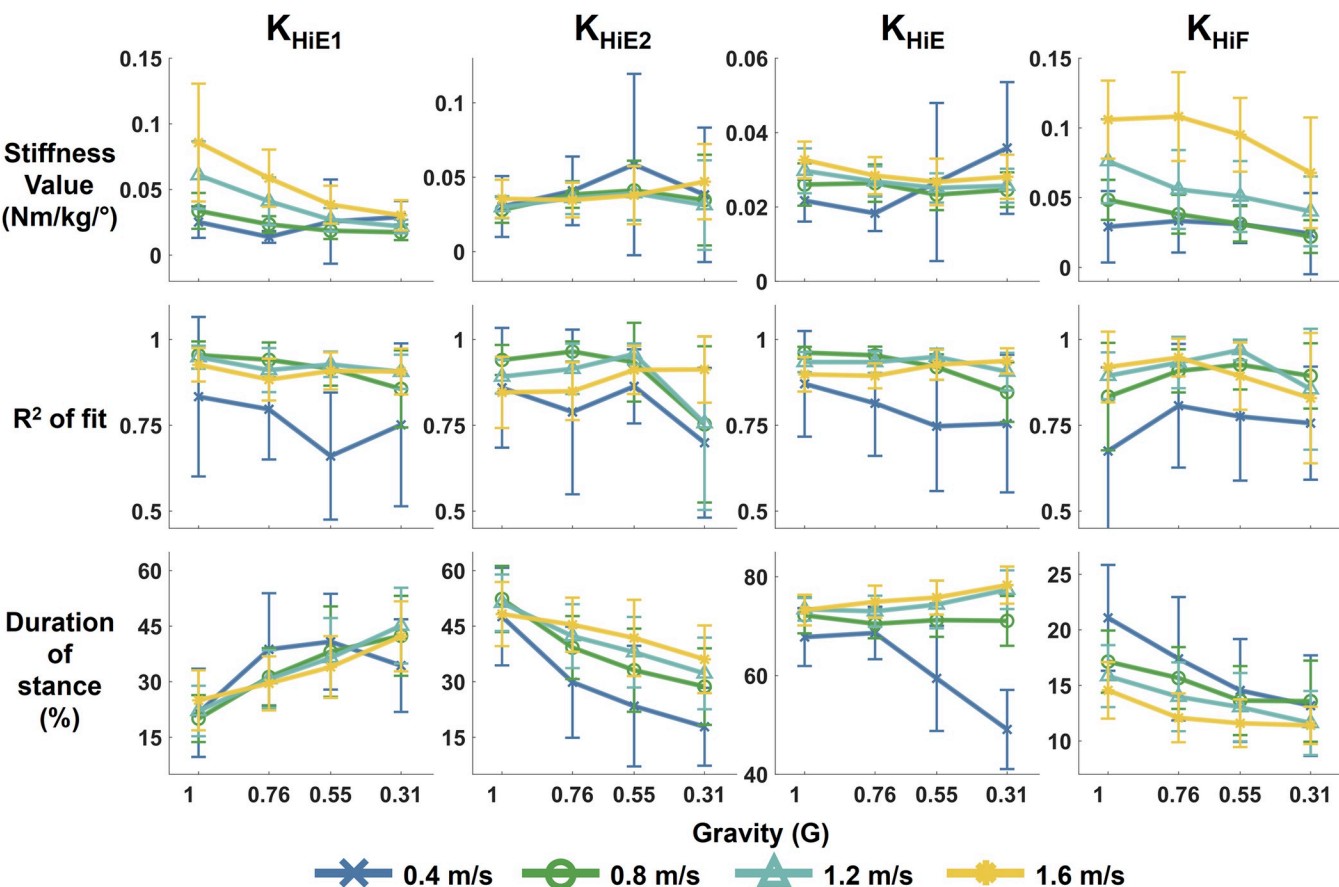

**Fig 7. Quasi-stiffness of the hip in stance phase at different levels of simulated reduced gravity and speeds.** Top row shows stiffness values, middle row is the quality of fit of the linear quasi-stiffness model, and bottom row is the duration of the quasi-stiffness phase as percent of stance phase. Each column of plots is a different quasi-stiffness phase for the hip: $K_{HiE1}$ is early hip extension, $K_{HiE2}$ is hip late extension, $K_{HiE}$ is overall hip extension, and $K_{HiF}$ is knee flexion. The error bars illustrate standard deviation.

units. Simulated reduced gravity reduced the load through the joints and moments about the leg joints, and reduced the muscle activation amplitudes during stance [21]. At lower muscle forces, there is less stretch of the tendon which moves the tendon stiffness into the short range stiffness known as the toe region. The toe region occurs at low levels of tendon stretch and incurs a smaller stiffness than in the subsequent linear phase of tendon length-force relationship. However, an alternative possibility is that subjects actively adjusted their muscle activation patterns to achieve close to normal joint displacements at reduced gravity. This approach would also produce a reduced joint quasi-stiffness at reduced gravity, completely independent of underlying muscle-tendon unit mechanical properties. Future research could use ultrasound imaging [45] to provide a better indication of intrinsic muscle-tendon displacements and stiffnesses during walking under simulated reduced gravity.

Our implementation of a rigid foot model may have led to an underestimation of the true ankle quasi-stiffness. We modelled the foot as a single rigid body, assuming no rotation or translation within the joints of the foot. Investigation of ankle quasi-stiffness in hopping found that modelling the ankle and foot separately, allowing for deformation of the arch of the foot, resulted in a substantially higher ankle quasi-stiffness than when the foot was modelled as a single rigid body [50].

## Ankle quasi-stiffness and selection of points of interest

Ankle quasi-stiffness decreased with reduced gravity but exhibited large variations at extreme speeds. The variation in $K_{AnD1}$ was very large during slow walking (0.4 m/s) although the variation decreased with reduced gravity. The individual trial data showed substantial variation in angle-moment loops in 0.4 m/s walking both within and between participants. This is reflected in large variations across bodyweight conditions for the $R^2$ fit and percent of stance spent in early dorsiflexion. We determined quasi-stiffness to be a good model for early-dorsiflexion ($K_{AnD1}$) and plantarflexion ($K_{AnPF}$) in all conditions ($R^2 > 0.74$ and $R^2 > 0.79$ respectively). Our statistical results confirm that gravity level is a greater predictor of ankle quasi-stiffness value, but that across quasi-stiffness phases, speed is the most influential factor for $R^2$ fit.

Quasi-stiffness provided an acceptable approximation of the moment-angle relationship in late-dorsiflexion ($K_{AnD2}$) and total dorsiflexion ($K_{AnDF}$) phases, but the quality of fit was decreased with reduced gravity and faster walking speeds ($R^2 \sim = 0.5$ at 1.6 m/s for $K_{AnD2}$ and $K_{AnDF}$ in 0.55G and 0.31 G respectively). The poor linearity of the late and total dorsiflexion phases at 1.6 m/s (and 1.2 m/s to a lesser extent) was related to the peak plantarflexion moment occurring after peak dorsiflexion angle. In normal gravity walking, the ankle angle continued to increase in dorsiflexion until the peak plantarflexion moment was reached, at which point the ankle dorsiflexion angle began to decrease. In fast and normal walking (1.6 & 1.2 m/s), particularly at low gravity, the moment-angle loop changed direction, with the ankle dorsiflexion angle decreasing before peak plantarflexion moment occurred. Therefore, the late dorsiflexion and total dorsiflexion phases exhibited non-linear behaviour and were not closely modelled by a linear model of quasi-stiffness.

We defined phases of quasi-stiffness and delimiting points of interest with reference to past literature and consideration of changes in moment-angle relationship across speeds and gravity levels. Previous studies used 4 points of interest to examine ankle quasi-stiffness in 3 distinct and approximately linear phases: early dorsiflexion, late dorsiflexion, and plantarflexion. The plantarflexion moment above a threshold [6,51,52], and minimum dorsiflexion angle [5] were used in separate studies to define point $P1_{An}$. We chose to use the minimum plantarflexion moment as the ankle angle profile changed substantially with gravity. In some low gravity conditions, the minimum dorsiflexion angle occurred close to the temporal mid-point of stance and no local minimum existed close to the beginning of stance. Although minimum ankle angle could have been a more suitable $P1_{An}$ at normal-fast walking speeds (1.2 & 1.6 m/s) in normal gravity, the point of minimum ankle plantarflexion moment was much more robust at capturing the onset of the linear early dorsiflexion phase of the ankle across all speeds and levels of simulated gravity. Our findings agreed with previous studies that determined $P3_{An}$ and $P4_{An}$ were suitably defined by maximum plantarflexion moment and toe-off respectively [5,7,19].

Whereas our definitions of $P1_{An}$, $P3_{An}$, and $P4_{An}$ were shared with previous works, we found that previously established definitions of $P2_{An}$ were not suitable due to the large variety of morphologies in moment-angle relationships across gravity and speed levels. Shamaei et al. and Krupenevich et al. identified the point of change between early and late dorsiflexion as ~30% of the gait cycle [5,17]. Crenna et al. used a 1.7x increase in the incremental ratio between moment and angle (the local slope of the relationship) to identify the transition between early and late dorsiflexion [6]. We initially considered the local minimum of vertical ground reaction force to determine $P2_{An}$ but found no local minimum in slow walking (0.4 m/s) and in medium walking (0.8 m/s) with high levels of reduced gravity. The wide range of moment-angle relationships meant that there was not an obvious change in slope or quasi-stiffness during the dorsiflexion stage for some walking conditions. For the reasons stated

prior, we chose to define the transition between early and late dorsiflexion as the temporal mid-point (50%) of stance-phase. At normal and fast walking speeds (1.2 & 1.6 m/s) in gravity levels close to normal, the mid-point of stance coincided with the local minimum ground reaction force and visual change in moment-angle relationship. At slow-medium speeds (0.4 & 0.8 m/s) and medium levels of gravity, the mid-point of stance was able to consistently partition the dorsi-flexion phase. At low gravity and slow-medium speeds (0.4 & 0.8 m/s), the temporal mid-point of stance occasionally occurred very close to or after the peak moment. In these circumstances, initial and/or later dorsiflexion quasi-stiffness were not applicable and dorsi-flexion was only modelled as a single linear phase ($K_{AnDF}$). Previous studies have evaluated quasi-stiffness over the entire period of dorsi-flexion, making a linear approximation of the often non-linear phase of dorsiflexion [2,19]. Benefits to modelling quasi-stiffness for the entire dorsiflexion phase, instead of 2 discrete phases, include model simplicity and ability to identify the start and end of the dorsiflexion phase in real time. Because $P2_{An}$ was dependant on stance duration, it could only be found in post-processing or approximated in real-time.

## Knee quasi-stiffness and selection of points of interest

Our model of knee joint quasi-stiffness was in close agreement with previous studies which considered quasi-stiffness for the knee flexion and extension phases occurring in early and mid-stance [1,4]. All 3 points of interest ($P1_{Kn}, P2_{Kn}, and P3_{Kn}$) were easily identifiable across levels of simulated gravity and speeds. The quasi-stiffness during knee flexion ($K_{KnF}$) significantly reduced with gravity level (P<0.001), but the quasi-stiffness of the extension stage ($K_{KnE}$) was not affected (P>0.99). $K_{KnF}$ exhibited a $R^2 > 0.85$ across all gravity and speed levels, indicating that quasi-stiffness was a good model of the knee during power absorption. In addition to a smaller peak knee extension moment, reducing gravity reduced the peak knee extension angle in early-mid stance. A smaller extension moment in combination with a more flexed knee created a rounded moment-angle profile, thus reducing linearity of the extension phase and ability of the quasi-stiffness model to accurately capture knee dynamics. In the lowest gravity condition, $R^2$ of $K_{KnE}$ was less than 0.7. $K_{KnE}$ also had a poor fit ($R^2 \sim = 0.5$) at 0.4 m/s in normal gravity. We therefore concluded that $K_{KnE}$ was a suitable model in normal (1 G) to 0.54 G at 0.8–1.6 m/s, but was unsuitable in the lowest gravity level (0.31 G) across speeds and for 0.4 m/s walking speed in all gravity levels. Previous studies averaged $K_{KnF}$ and $K_{KnE}$ to find the knee quasi-stiffness throughout early-mid stance [4]. However, a single quasi-stiffness model of the knee would be unsuitable and inaccurate at low gravities due to the widening of the moment-angle loop.

The duration spent in the knee flexion and extension phase typically only accounted for ~60% of the stance phase, suggesting that a substantial proportion of the knee dynamics are not modelled by our quasi-stiffness parameters. Future studies on knee joint quasi-stiffness may choose to consider a 3rd phase of quasi stiffness between local minimum knee extension moment ($P3_{Kn}$) and the local maxima of knee extension before toe-off. We identified a local extension moment maxima at ~80% of stance in all speeds and gravity conditions. Linear quasi-stiffness may be a suitable model for this extension phase and a 4th phase of quasi-stiffness could also approximate the moment-angle relationship between the local extension moment maxima and toe-off.

## Hip quasi-stiffness and selection of points of interest

Quasi-stiffness provided a good model of hip dynamics throughout stance and we found the quasi-stiffness of the early extension ($K_{HiE1}$) and flexion ($K_{HiF}$) phases were dependant on gravity level (p = 0.004 and p<0.001 respectively). Although $K_{HiF}$ decreased with gravity, the

percent of stance spent in the flexion phase also decreased with gravity. At each walking speed in the lowest gravity condition, at least 2 subjects did not exhibit a hip flexion phase, with the proportion of subjects increasing with walking speed (S2 Table). The lack of hip flexion at low gravity across speeds was further evidence of the subjects adopting different gait strategies when ground reaction forces are reduced. In normal gravity, $K_{HiF}$ closely approximates hip moment-angle relationship, except for 0.4 m/s walking. Therefore, $K_{HiF}$ was a suitable model for gravity levels 0.54, 0.76, and 1 G for walking speeds between 0.8 and 1.6 m/s.

In general, the $R^2$ fit of each phase of hip quasi-stiffness was good across all levels of gravity, but 0.4 m/s had both the lowest $R^2$ values and the largest variation in standard deviation. Due to the large variation in mean $R^2$ at 0.4 m/s, we found that quasi-stiffness was not a suitable model for walking at slow speeds (0.4 m/s). Quasi-stiffness control is therefore not a suitable assistance strategy for wearable devices to assist slow walkers.

We developed a new hip quasi-stiffness model with reference to two previously established models. An earlier model of hip quasi-stiffness, developed by Frigo et al., identified 4 phases of quasi-stiffness for the hip: 3 phases of extension in stance phase separated by different slopes, and 1 phase of extension during the swing phase [1]. Shamaei et al. modelled an extension and flexion phase of quasi-stiffness using maximum hip flexion moment as the transition point between them and local deflection points as the onset of the extension and termination of the flexion phase [3]. The local deflection was defined as the point where the double derivative of moment with respect to angle showed a local discontinuity. To ensure our model was robust to a wide range of gait speeds and gravity levels, we chose to model 2 distinct phases of extension, an overall extension phase, and a flexion stage. Our model agreed with previous studies in that the peak hip flexion moment defined the termination of the extension stage. Our delimiting point between the 2 phases of extension was the transition of hip moment from extension to flexion as this metric was relatively easy to measure in real time, and could be found across almost all gravity levels and gait speeds. Our $R^2$ results showed that the extension phases were able to reasonably approximate hip quasi-stiffness. Unique to our model, we calculated $K_{HiE}$ (from minimum to maximum hip flexion moment) and found this to be a good model ($R^2 > 0.74$) of the moment-angle relationship at speeds greater than 0.4 m/s. This single quasi-stiffness value was able to model hip dynamics for up to 80% of stance across gravity levels. We were unable to calculate early and late extension stages for some walking conditions as the hip moment remained in extension for the duration of stance. The single quasi-stiffness model of the hip extension phase is a more robust model than a 2-phase extension model of hip dynamics at a wide range of speeds and gravity levels.

## Implications of findings

Our quasi-stiffness models of the ankle, knee, and hip have implications for robotic control of finite-state machine controlled bipedal robots, prosthetics, and exoskeletons. Exoskeletons and prosthetics can emulate quasi-stiffness during a phase of gait by using finite-state machine controllers to identify phases of gait and modify effective stiffness of the robotic joint [10,53–57]. All of the current biomimetic exoskeletons and prostheses that assist gait by adjusting device quasi-stiffness are designed for walking at normal gravity. Our findings suggest that similar devices with reduced stiffness values would be able to assist gait in reduced gravity overground walking, which is a therapy often used for gait rehabilitation. We also identified specific linear phases in the joint moment-angle relationships during stance where robotic quasi-stiffness assistance would be beneficial and reported the biological measurements that indicate beginning and end of these phases.

Gravity level and joint quasi-stiffness had a linear but non-proportional relationship. Due to the complexity of the relationship and the interplay of walking speed, the quasi-stiffness of a leg joint at normal gravity cannot be multiplied by the gravity level to predict quasi-stiffness of the joint in reduced gravity.

Passive or quasi-passive exoskeletons use springs to assist locomotion and our results could be valuable in determining optimal spring stiffnesses and period of engagement. Passive running ankle prostheses most-commonly use a spring-leaf spring with specific stiffness to aid in ankle plantarflexion energy storage and return [58,59]. Some knee prostheses also use quasi-passive mechanisms to assist gait [60,61]. Passive or quasi-passive exoskeletons also use passive stretch and relax of springs in parallel with the legs to assist gait [62–67]. Our results showed that 6 phases of quasi-stiffness were significantly decreased by gravity level and therefore a single spring stiffness would not be optimal for multiple gravity levels. Previous research found that walking speed impacts quasi-stiffness of joints [51,52,68] and mechanisms have been developed to vary spring stiffness across gaits. We conclude that similar adjustments are needed for walking in low gravity.

Quasi-stiffness of leg joints in reduced gravity may also have design implications for extra-vehicular space suits. Space suits are pressurized, which makes the suit stiff. Increased suit stiffness makes it harder or more energetically expensive to bend leg joints as the joint must overcome the stiffness imposed by the suit. Integration of exoskeletons in space-suits could assist movement in low-gravity environments like Mars or the Moon [69,70]. The integrated exoskeleton could be designed to assist walking by mimicking the biological joint quasi stiffness at low gravity in non-pressurized conditions. Our results detail the relationship between gravity and joint-quasi stiffness and could be used to design a gait assisting exoskeleton integrated into a space suit.

## Limitations

There were a few limitations to our study. The first is that we controlled for speed using the average speed over a 4 m section of the walk-way, centered around the forceplates. Participants completed 1–3 strides during the period we controlled for speed and their gait speed may have varied at different points of the walkway. We asked participants to maintain an even walking pace throughout the walkway and post-processing confirmed that walking speeds were close to the target walking speed. It should be noticed that participants struggled to walk very slowly (0.4 m/s) in normal gravity. We chose to include 0.4 m/s as a testing condition because people with walking difficulties, who would benefit from walking with an exoskeleton in reduced gravity environments, have a slower preferred and maximal speed. It is also worth noting that we tested with only healthy-able bodied participants. Joint quasi-stiffness of people with physical limitations may be different to that of able-bodied people and could react differently in response to altered gravity. If the neuromuscular control of quasi-stiffness was a property of inherent muscle stiffness as supported by our findings, then it would stand to reason that joint quasi-stiffness of people with gait disabilities would also decrease with gravity. The statistical power of our results could have been improved with more subjects. The number of subjects in this study is comparable to previous studies on bodyweight supported walking or joint quasi-stiffness [38,39].

## Conclusion

Quasi-stiffness tended to decrease with simulated reduced gravity and supported the theory that joint quasi-stiffness is determined by the inherent stiffness properties of the muscle-tendon units. We identified 4 phases of quasi-stiffness in both the ankle and hip, and 2 phases of

quasi-stiffness in the knee that approximated moment-angle loops across different levels of gravity and walking speeds. At low levels of gravity, the relative power and strength of the joints were increased and subjects could choose from a wide selection of gait strategies. Our findings on quasi-stiffness in reduced gravity have implications for passive, quasi-passive, and finite-state machine controlled prostheses, and exoskeletons.

## Supporting information

**S1 Table. Participant descriptive data.**
(DOCX)

**S2 Table. Walking conditions and number of subjects for which phases of quasi-stiffness could not be calculated.** Numbers in bold indicate conditions where data were lost.
(DOCX)

**S3 Table. Mean quasi-stiffness values with standard deviations for all conditions.**
(DOCX)

**S4 Table. Mean $R^2$ for each quasi-stiffness linear model with standard deviations for all conditions.**
(DOCX)

## Acknowledgments

We thanks the assistance of Angel Bu, Han Nguyen, Patrick Costello, and Skyler Levine in collecting and cleaning data.

## Author Contributions

**Conceptualization:** Mhairi K. MacLean, Daniel P. Ferris.

**Formal analysis:** Mhairi K. MacLean.

**Funding acquisition:** Daniel P. Ferris.

**Investigation:** Mhairi K. MacLean.

**Methodology:** Mhairi K. MacLean, Daniel P. Ferris.

**Project administration:** Mhairi K. MacLean, Daniel P. Ferris.

**Resources:** Daniel P. Ferris.

**Supervision:** Daniel P. Ferris.

**Visualization:** Mhairi K. MacLean.

**Writing – original draft:** Mhairi K. MacLean.

**Writing – review & editing:** Mhairi K. MacLean, Daniel P. Ferris.

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
