## [Decision Letter · Decision Letter 0]

31 Mar 2022

PONE-D-21-38355Effects of simulated reduced gravity and walking speed on ankle, knee, and hip quasi-stiffness in overground walkingPLOS ONE

Dear Dr. MacLean,

Thank you for submitting your manuscript to PLOS ONE. After careful consideration, we feel that it has merit but does not fully meet PLOS ONE’s publication criteria as it currently stands. Therefore, we invite you to submit a revised version of the manuscript that addresses the points raised during the review process.

We look forward to receiving your revised manuscript.

Kind regards,

J. Lucas McKay, Ph.D., M.S.C.R.

Academic Editor

PLOS ONE

Journal Requirements:

Reviewers' comments:

Reviewer's Responses to Questions

**Comments to the Author**

1. Is the manuscript technically sound, and do the data support the conclusions?

Reviewer #1: Yes

Reviewer #2: Yes

2. Has the statistical analysis been performed appropriately and rigorously? 

Reviewer #1: Yes

Reviewer #2: Yes

3. Have the authors made all data underlying the findings in their manuscript fully available?

Reviewer #1: Yes

Reviewer #2: Yes

4. Is the manuscript presented in an intelligible fashion and written in standard English?

Reviewer #1: Yes

Reviewer #2: Yes

5. Review Comments to the Author

Reviewer #1: This study aims at investigating the effect of simulated reduced gravity (provided by a bodyweight support system) and speed on quasi-stiffness of the hip, knee and ankle during overground walking. Quantifying quasi-stiffness during simulated reduced gravity may have important implications for the design of prosthetics and exoskeletons when external bodyweight support is provided. The authors found that quasi-stiffness decrease with simulated reduced gravity and speed is a main factor in determining quasi-stiffness. They concluded that joint quasi-stiffness is determined by the inherent stiffness properties of the muscle tendon units. The manuscript is well written and the quality of the data, analysis and figures provide enough evidence to support the conclusions.

Minor comment:

The proposed hypothesis (“We hypothesized that joint quasi-stiffness would decrease with simulated reduced gravity as it is likely dependent on peak muscle and tendon force”) is a prediction of the results given the relationship between joint net moments and quasi-stiffness. I would suggest to reformulating the hypothesis in relation to the proposed biomechanical principles underlying the modulation of quasi-stiffness.

Reviewer #2: The current paper aimed to determine how joint quasi-stiffness changes in response to different levels of simulated reduced gravity at a range of walking speeds. Quasi-stiffness is a useful tool for controlling robotic devices and has yet to be studied at different gravity levels. Twelve healthy, young participants completed the protocol, which involved overground walking over a force plate at prescribed speeds and gravity levels. The results support the authors’ hypothesis that quasi-stiffness would decrease with reductions in simulated gravity. The discussion compares and contrasts the quasi-stiffness parameters to that of previous literature and proposes explanations for the mechanism behind the changes in quasi-stiffness.

While the paper is well-written, there are a handful of typos and omissions (some of which I have pointed out below). In addition, I have suggested some more impactful revisions pertaining to clarifying the methodology and focusing the intro and discussion.

The most important revisions deal with ensuring that the framing of the conclusions is appropriate for the scope of this study.

Intro:

Throughout the intro, please include citations to back up statements of known phenomena. For example, page 9, line 23-24, multiple studies have shown a decrease in GRF due to simulated reduced gravity many of which are referenced elsewhere in this manuscript.

The section on simulated reduced gravity as a gait rehabilitation therapy is lacking a clear, concise message. In the current study, gait rehabilitation appears to have motivated the use of the slowest walking speed, and in the discussion the idea of rehabilitative devices is touched on. As is, the paragraph does not appear to add meaningfully to the introduction, other than to state that body-weight support is used as a therapeutic tool. Perhaps joint quasi-stiffness has been studied in these populations and could be added to this section?

Method:

Page 12, lines 5-7: I think this sentence is incomplete, and may be intended to provide references for the quasi-stiffness phases, which would be helpful.

Figure 1: These figures help quite a bit with visualizing and understanding the quasi-stiffness calculation and results. However, there are quite a few elements and acronyms, which make the figure difficult to understand quickly: it needs to be studied while referencing Tables 1 & 2. I recommend adding a figure that shows joint moment over time and indicates the phases over which quasi-stiffness was calculated, which will allow readers to more easily understand when the points of interest occur in the gait cycle

Page 13, line 5: I assume ‘mid-point of stance’ refers to the temporal mid-point. Is this correct? Or is it based on a spatial metric? Please clarify.

Page 13, line 5: planter > plantar

Page 14, line12: Please specify which speeds you mean by ‘normal’.

Page 14, line 22: Please add some more specifics to this section. It is not clear when the analysis is being run on a single stride, single condition, single subject, etc. For example, as written, it seems that you calculated quasi-stiffness using a least squares model for a given phase within a single stride. Then you used the resulting model to predict the joint moment from the joint angle for that same phase. It is not clear to me how that is different from the R^2 of the linear model, unless different strides or subjects were compared.

Results:

Figure 3: I recommend swapping the positions of the Kanpf column and Kandf column, so that the columns are more chronological from left to right. Also, the data labels for each speed level (x vs o vs triangle etc) are very clear in the legend, but not clear in the figure itself, which may make this figure hard to read for a person with color-blindness. Please make the data labels more visible. Lastly, please indicate in the figure caption which measure of error the error bars are indicating (likely standard deviation).

Page 17, line 9: This appears to be the first and only use of these acronyms, so please state with words instead of acronyms.

Figure 5: Similar to Figure 3, I recommend swapping the positions of Khif and Khie.

Discussion:

Section: Biomechanical Principles Underlying Joint Quasi-Stiffness

While I agree with the sentiment of this section, I think it should be slightly reframed. The conclusion that ‘…quasi-stiffness is modulated by the inherent properties of the muscle-tendon units” is perhaps a stretch beyond the scope of the current study. The section could be strengthened by including discussion on how the interaction between tendon and muscle dynamics are impacted by different loads and how these changes could contribute to explaining the results of the current study. Additionally, I was unable to find/access a complete reference for citation 41, which appears to be important for this argument.

Page 20, line 6 & throughout: The authors make several references to ‘normal’ walking speed. There are four speeds, and it is unclear which speed or speeds they are referring to as normal. Please be more specific when referencing results. One solution would be to use a parenthetical such as ‘…normal walking (1.2 m/s)…’ or to define what constitutes ‘normal’ at first mention.

Page 21, line 14: ‘liner’ > ‘linear’

Page 22, line 23: Could the change in hip kinematics be related to how the participants are interacting with the body-weight support system? According to the 2020 Maclean paper, slower speeds saw higher forward pulling forces. Perhaps it would be worthwhile to investigate trunk angle to determine how the lack of hip flexion could come about.

Page 23, line 2: ‘worst’ > ‘lowest’

Page 24, line 10: ‘liner’ > ‘linear’

6. PLOS authors have the option to publish the peer review history of their article (what does this mean?). If published, this will include your full peer review and any attached files.

Reviewer #1: No

Reviewer #2: No

---

## [Author Response · Author response to Decision Letter 0]

31 May 2022

Reviewer #1: 

This study aims at investigating the effect of simulated reduced gravity (provided by a bodyweight support system) and speed on quasi-stiffness of the hip, knee and ankle during overground walking. Quantifying quasi-stiffness during simulated reduced gravity may have important implications for the design of prosthetics and exoskeletons when external bodyweight support is provided. The authors found that quasi-stiffness decrease with simulated reduced gravity and speed is a main factor in determining quasi-stiffness. They concluded that joint quasi-stiffness is determined by the inherent stiffness properties of the muscle tendon units. The manuscript is well written and the quality of the data, analysis and figures provide enough evidence to support the conclusions.

We thank the reviewer for their feedback and are happy to hear that the paper is well-written. 

Minor comment:

The proposed hypothesis (“We hypothesized that joint quasi-stiffness would decrease with simulated reduced gravity as it is likely dependent on peak muscle and tendon force”) is a prediction of the results given the relationship between joint net moments and quasi-stiffness. I would suggest to reformulating the hypothesis in relation to the proposed biomechanical principles underlying the modulation of quasi-stiffness.

Hypothesis has been modified to include the intrinsic muscle-tendon unit stiffness and some references to back up our hypothesis. 

“We hypothesized that joint quasi-stiffness would decrease with simulated reduced gravity as it is likely dependent on the intrinsic muscle-tendon unit stiffness which scales with muscle and tendon force [15,43–45].” 

Reviewer #2: 

The current paper aimed to determine how joint quasi-stiffness changes in response to different levels of simulated reduced gravity at a range of walking speeds. Quasi-stiffness is a useful tool for controlling robotic devices and has yet to be studied at different gravity levels. Twelve healthy, young participants completed the protocol, which involved overground walking over a force plate at prescribed speeds and gravity levels. The results support the authors’ hypothesis that quasi-stiffness would decrease with reductions in simulated gravity. The discussion compares and contrasts the quasi-stiffness parameters to that of previous literature and proposes explanations for the mechanism behind the changes in quasi-stiffness.

While the paper is well-written, there are a handful of typos and omissions (some of which I have pointed out below). In addition, I have suggested some more impactful revisions pertaining to clarifying the methodology and focusing the intro and discussion.

The most important revisions deal with ensuring that the framing of the conclusions is appropriate for the scope of this study.

We thank the reviewer for their detailed feedback and suggestions to improve manuscript quality. We’re also happy to hear that the review found the paper well-written. We have endeavoured to incorporate your feedback to improve the manuscript.

Intro:

Throughout the intro, please include citations to back up statements of known phenomena. For example, page 9, line 23-24, multiple studies have shown a decrease in GRF due to simulated reduced gravity many of which are referenced elsewhere in this manuscript.

We have added some references to back up statements of known phenomena. The sentences with new references in the introduction are:

“Study of joint quasi-stiffness during different gait conditions may also provide insight into human neural control [13,14].”

“Joint quasi-stiffness is determined by the state of all passive and active tissues that cross the joint [15,16].”

“Used as a gait rehabilitation therapy, simulated reduced gravity reduces the joint torques and muscle forces required to produce gait [20,21].” 

“One technique to simulate reduced gravity is to apply a constant upwards force to the torso with a harness, counteracting gravity without altering body mass [22].”

“Reducing the effective bodyweight also decreases joint contact forces and net muscle moments during walking, essentially making it easier to walk [20,21,26].”

“We hypothesized that joint quasi-stiffness would decrease with simulated reduced gravity as it is likely dependent on the intrinsic muscle-tendon unit stiffness which scales with muscle and tendon force [15,43–45].”

The references used are: 

13. Wind AM, Rouse EJ. Neuromotor Regulation of Ankle Stiffness is Comparable to Regulation of Joint Position and Torque at Moderate Levels. Sci Rep. 2020;10. doi:10.1038/S41598-020-67135-X

14. Rouse EJ, Hargrove LJ, Perreault EJ, Kuiken TA. Estimation of Human Ankle Impedance During the Stance Phase of Walking. IEEE Trans Neural Syst Rehabil Eng. 2014;22: 870. doi:10.1109/TNSRE.2014.2307256

15. Whittington B, Silder A, Heiderscheit B, Thelen DG. The contribution of passive-elastic mechanisms to lower extremity joint kinetics during human walking. Gait Posture. 2008;27: 628–634. doi:10.1016/J.GAITPOST.2007.08.005

16. Silder A, Heiderscheit B, Thelen DG. Active and passive contributions to joint kinetics during walking in older adults. J Biomech. 2008;41: 1520–1527. doi:10.1016/J.JBIOMECH.2008.02.016

20. MacLean MK, Ferris DP. Human muscle activity and lower limb biomechanics of overground walking at varying levels of simulated reduced gravity and gait speeds. PLoS One. 2021;16. doi:10.1371/JOURNAL.PONE.0253467

21. Goldberg SR, Stanhope SJ. Sensitivity of joint moments to changes in walking speed and body-weight-support are interdependent and vary across joints. J Biomech. 2013;46: 1176–1183. doi:10.1016/J.JBIOMECH.2013.01.001

22. Farley CT, McMahon TA. Energetics of walking and running: insights from simulated reduced-gravity experiments. J Appl Physiol. 1992;73: 2709–2712. doi:10.1152/jappl.1992.73.6.2709

26. Apte S, Plooij M, Vallery H. Influence of body weight unloading on human gait characteristics: a systematic review. J Neuroeng Rehabil. 2018;15. doi:10.1186/s12984-018-0380-0

43. Rack PMH, Westbury DR. The short range stiffness of active mammalian muscle and its effect on mechanical properties. J Physiol. 1974;240: 331. doi:10.1113/JPHYSIOL.1974.SP010613

44. Walmsley B, Proske U. Comparison of stiffness of soleus and medial gastrocnemius muscles in cats. J Neurophysiol. 1981;46: 250–259. doi:10.1152/JN.1981.46.2.250

45. Krupenevich RL, Beck ON, Sawicki GS, Franz JR. Reduced Achilles Tendon Stiffness Disrupts Calf Muscle Neuromechanics in Elderly Gait. Gerontology. 2022;68: 241–251. doi:10.1159/000516910

The section on simulated reduced gravity as a gait rehabilitation therapy is lacking a clear, concise message. In the current study, gait rehabilitation appears to have motivated the use of the slowest walking speed, and in the discussion the idea of rehabilitative devices is touched on. As is, the paragraph does not appear to add meaningfully to the introduction, other than to state that body-weight support is used as a therapeutic tool. Perhaps joint quasi-stiffness has been studied in these populations and could be added to this section?

In this section, we introduce and discuss the concept of simulated reduced gravity. Specifically, we highlight that the technique can reduce joint torque, which links back to the previous paragraph of exploring quasi-stiffness under such conditions. We think it is important to dedicate some of the introduction to an overview of bodyweight supported walking. To the best of our knowledge, joint quasi-stiffness has not been studied in a clinical population. To improve the relevancy of the discussion of gait rehabilitation, we have explained the benefit of understanding the effect of gravity level in a healthy population as it pertains to understanding if pathological gait has an interaction with gravity level. 

“To the best of our knowledge, there has been no investigation of joint quasi-stiffness in clinical populations in normal or reduced gravity. To parse out the effects and interactions of gravity level, walking speed, and underlying pathological physiology on quasi-stiffness, we first need an understanding of the relationship between gravity level, walking speed, and joint quasi-stiffness in a healthy population.”

Method:

Page 12, lines 5-7: I think this sentence is incomplete, and may be intended to provide references for the quasi-stiffness phases, which would be helpful.

This sentence was incomplete, thank you for catching that. We have completed it and added the 5 references relevant to building our quasi-stiffness models.

We chose phases of quasi-stiffness delimited by points of interest (Table 2) with reference to previous studies [1–6,17].

Figure 1: These figures help quite a bit with visualizing and understanding the quasi-stiffness calculation and results. However, there are quite a few elements and acronyms, which make the figure difficult to understand quickly: it needs to be studied while referencing Tables 1 & 2. I recommend adding a figure that shows joint moment over time and indicates the phases over which quasi-stiffness was calculated, which will allow readers to more easily understand when the points of interest occur in the gait cycle

We agree that there’s a lot of information on these figures and they required some time to be fully understood. We have endeavoured to improve clarity and split the figure into 3, with each joint moment-angle loop accompanied by the moment and angle during stance. We hope the consistency in colours and labelling between the figures, will help improve clarity. The text from each new figure is a small variation of the following:

“Figure X. Joint angle, moment, and moment-angle relationships during stance phase overlaid with points of interest and phases of quasi-stiffness. The smaller subplots on the left show joint angle and moment during stance phase. The dashed vertical lines represent the timings of each point of interest, while the open circles identify the point of interest itself. The phases of quasi-stiffness are identified at the bottom of these plots, with the arrows indicating their onset and termination. The large, right figure shows the moment-angle relationship with the quasi-stiffness overlayed as dashed lines and the points of interests as “+”. Data for visualization were normalized to 60 data points and averaged across all participants walking in normal gravity at 1.6 m/s. Direction of internal joint moment and joint angle are indicated with arrows on the rightmost plot.”

Page 13, line 5: I assume ‘mid-point of stance’ refers to the temporal mid-point. Is this correct? Or is it based on a spatial metric? Please clarify.

You assume correctly. We have clarified the mid-point is “temporal” at multiple points in the manuscript for clarity.

Page 14, line12: Please specify which speeds you mean by ‘normal’.

Page 20, line 6 & throughout: The authors make several references to ‘normal’ walking speed. There are four speeds, and it is unclear which speed or speeds they are referring to as normal. Please be more specific when referencing results. One solution would be to use a parenthetical such as ‘…normal walking (1.2 m/s)…’ or to define what constitutes ‘normal’ at first mention.

We agree that the speeds were somewhat confusing as they were originally were presented and have clarified walking speeds throughout the paper. Generally speaking, slow = 0.4 m/s, medium = 0.8 m/s, normal = 1.2 m/s, and fast = 1.6 m/s. 

Page 14, line 22: Please add some more specifics to this section. It is not clear when the analysis is being run on a single stride, single condition, single subject, etc. For example, as written, it seems that you calculated quasi-stiffness using a least squares model for a given phase within a single stride. Then you used the resulting model to predict the joint moment from the joint angle for that same phase. It is not clear to me how that is different from the R^2 of the linear model, unless different strides or subjects were compared.

We appreciate ethe feedback and have added additional information for clarity. In short, for every walking trial, the linear model was used to predict moment from angle and we found the R^2 for that trial. Stiffness and R^2 values were then averaged within conditions for each participants, and then averaged across participants. 

“We found the 11 points of interest for each walking trial, then calculated the stiffness of each quasi-stiffness phase using a least squares linear model. For each walking trial, we applied the linear model as a function of joint angle (i.e. predicted joint moment from given joint angles). To determine the fit of the model for each phase of quasi-stiffness in each trial, we calculated the R2 value. We also calculated the duration of each phase of quasi-stiffness as a percentage of the total stance time. Finally, we averaged quasi-stiffness values, R2, and phase durations across each walking trial in a condition for every participant, and then averaged across participants.”

Results:

Figure 3: I recommend swapping the positions of the Kanpf column and Kandf column, so that the columns are more chronological from left to right. Also, the data labels for each speed level (x vs o vs triangle etc) are very clear in the legend, but not clear in the figure itself, which may make this figure hard to read for a person with color-blindness. Please make the data labels more visible. Lastly, please indicate in the figure caption which measure of error the error bars are indicating (likely standard deviation).

We have reordered the columns to be ordered as suggested. They were originally ordered in strict chronological order, as Kandf does not follow on from Kand2, but we can see the benefit to grouping all the dorsi-flexion QS together and have implemented this in the figures and tables.

We chose the colours such that they would be discernible in greyscale and for several colour-blindness types. We used an online tool (link to tool with our colours pre-loaded) to visualize the colours for different colour-blindness types and converted the figures to greyscale to check for clarity. The 0.4 and 0.8 m/s colours are the closest in greyscale, so we chose the label markers that are most visually different (cross and open circle). Unfortunately, increasing the size of the markers makes the plots too noisy and the data more difficult to read. We have included that the error bars represent standard deviation. The new figure caption reads:

“Figure 5. Quasi-stiffness of the ankle in stance phase at different levels of simulated reduced gravity and speeds. Top row shows stiffness values, middle row is the quality of fit of the linear quasi-stiffness model, and bottom row is the duration of the quasi-stiffness phase as percent of stance phase. Each column of plots is a different quasi-stiffness phase for the ankle: KAnD1 is early dorsi-flexion, KAnD2 is Ankle late dorsi-flexion, KAnDF is overall Ankle dorsi-flexion, and KAnPF is Ankle plantar-flexion. The error bars represent standard deviation.”

Page 17, line 9: This appears to be the first and only use of these acronyms, so please state with words instead of acronyms.

Thank you for bringing our attention to this! The PKn, QKn, and RKn acronyms were accidental hold-outs of an older set of acronyms used when drafting the manuscript. These acronyms have been updated to P1Kn, P2Kn, and P3Kn respectively. The text in the manuscript reads as:

“There were no instances where P1Kn, P2Kn, or P3Kn could not be identified.”

Figure 5: Similar to Figure 3, I recommend swapping the positions of Khif and Khie.

We have swapped the order of the appearance for this figure and relevant table. The final 2 sentences of the figure caption reads:

“Each column of plots is a different quasi-stiffness phase for the hip: KHiE1 is early hip extension, KHiE2 is hip late extension, KHiE is overall hip extension, and KHiF is knee flexion. The error bars illustrate standard deviation.”

Discussion:

Section: Biomechanical Principles Underlying Joint Quasi-Stiffness

While I agree with the sentiment of this section, I think it should be slightly reframed. The conclusion that ‘…quasi-stiffness is modulated by the inherent properties of the muscle-tendon units” is perhaps a stretch beyond the scope of the current study. The section could be strengthened by including discussion on how the interaction between tendon and muscle dynamics are impacted by different loads and how these changes could contribute to explaining the results of the current study. Additionally, I was unable to find/access a complete reference for citation 41, which appears to be important for this argument.

We have softened this section so our discussion stays within the scope of this study. Citation 41 is still in review, so we have removed it from this manuscript. The section now reads as:

“Joint quasi-stiffness tended to decrease with gravity level, which is in agreement with the supposition that quasi-stiffness is modulated by the inherent properties of the muscle-tendon units. Simulated reduced gravity reduced the load through the joints and moments about the leg joints, and reduced the muscle activation amplitudes during stance [21]. At lower muscle forces, there is less stretch of the tendon which moves the tendon stiffness into the short range stiffness known as the toe region. The toe region occurs at low levels of tendon stretch and incurs a smaller stiffness than in the subsequent linear phase of tendon length-force relationship. However, an alternative possibility is that subjects actively adjusted their muscle activation patterns to achieve close to normal joint displacements at reduced gravity. This approach would also produce a reduced joint quasi-stiffness at reduced gravity, completely independent of underlying muscle-tendon unit mechanical properties. Future research could use ultrasound imaging [45] to provide a better indication of intrinsic muscle-tendon displacements and stiffnesses during walking under simulated reduced gravity.

“Our implementation of a rigid foot model may have led to an underestimation of the true ankle quasi-stiffness. We modelled the foot as a single rigid body, assuming no rotation or translation within the joints of the foot. Previous research has found that modelling the ankle and foot separately, allowing for deformation of the arch of the foot, underestimated quasi-stiffness of the ankle in hopping [50]. “

Page 22, line 23: Could the change in hip kinematics be related to how the participants are interacting with the body-weight support system? According to the 2020 Maclean paper, slower speeds saw higher forward pulling forces. Perhaps it would be worthwhile to investigate trunk angle to determine how the lack of hip flexion could come about.

Thank-you for your insights. We do not think the pulling forces would have impacted the lack of hip flexion. The pulling forces at all gravity levels rarely exceeded 1% of the person’s bodyweight (in normal gravity). We suggest the lack of hip flexion is an adoption of a new walking strategy. We did not have any markers above the hips, so we cannot retroactively examine trunk angle. 

Page 13, line 5: planter > plantar

Page 21, line 14: ‘liner’ > ‘linear’

Page 24, line 10: ‘liner’ > ‘linear’

Thank you for pointing out these spelling errors. We have corrected them in the revision.

Page 23, line 2: ‘worst’ > ‘lowest’

We agree that ‘lowest’ is a better choice than ‘worst’ and have updated the word choice in the revision. 

We have tracked changes in the manuscript revision, and highlighted yellow any changes we made which could not be tracked (i.e. referencing or table formatting)

---

## [Decision Letter · Decision Letter 1]

11 Jul 2022

Effects of simulated reduced gravity and walking speed on ankle, knee, and hip quasi-stiffness in overground walking

PONE-D-21-38355R1

Dear Dr. MacLean,

We’re pleased to inform you that your manuscript has been judged scientifically suitable for publication and will be formally accepted for publication once it meets all outstanding technical requirements.

Kind regards,

Kei Masani

Academic Editor

PLOS ONE

Additional Editor Comments (optional):

Reviewers' comments:

Reviewer's Responses to Questions

**Comments to the Author**

1. If the authors have adequately addressed your comments raised in a previous round of review and you feel that this manuscript is now acceptable for publication, you may indicate that here to bypass the “Comments to the Author” section, enter your conflict of interest statement in the “Confidential to Editor” section, and submit your "Accept" recommendation.

Reviewer #1: All comments have been addressed

Reviewer #2: All comments have been addressed

2. Is the manuscript technically sound, and do the data support the conclusions?

Reviewer #1: Yes

Reviewer #2: Yes

3. Has the statistical analysis been performed appropriately and rigorously? 

Reviewer #1: Yes

Reviewer #2: Yes

4. Have the authors made all data underlying the findings in their manuscript fully available?

Reviewer #1: Yes

Reviewer #2: Yes

5. Is the manuscript presented in an intelligible fashion and written in standard English?

Reviewer #1: Yes

Reviewer #2: Yes

6. Review Comments to the Author

Reviewer #1: (No Response)

Reviewer #2: The authors have done an adequate job addressing all previous concerns, resulting in a nice manuscript describing a well performed study.

7. PLOS authors have the option to publish the peer review history of their article (what does this mean?). If published, this will include your full peer review and any attached files.

Reviewer #1: No

Reviewer #2: No

---

## [Editor Report · Acceptance letter]

29 Jul 2022

PONE-D-21-38355R1 

Effects of simulated reduced gravity and walking speed on ankle, knee, and hip quasi-stiffness in overground walking 

Dear Dr. MacLean:

I'm pleased to inform you that your manuscript has been deemed suitable for publication in PLOS ONE. Congratulations! Your manuscript is now with our production department. 

Kind regards, 

on behalf of

Dr. Kei Masani 

Academic Editor

PLOS ONE